# Integrating polygenic risk scores in the prediction of type 2 diabetes risk and subtypes in British Pakistanis and Bangladeshis: A population-based cohort study

Sam Hodgson[1☉], Qin Qin Huang[2☉], Neneh Sallah[3,4], Genes & Health Research Team[5,6¶], Chris J. Griffiths[5], William G. Newman[7,8], Richard C. Trembath[9], John Wright[10], R. Thomas Lumbers[3,11], Karoline Kuchenbaecker[4,12], David A. van Heel[6], Rohini Mathur[13], Hilary C. Martin[2☉], Sarah Finer[5☉]*

1 Primary Care Research Centre, University of Southampton, Southampton, United Kingdom, 2 Department of Human Genetics, Wellcome Sanger Institute, Hinxton, United Kingdom, 3 Institute of Health Informatics, University College London, London, United Kingdom, 4 UCL Genetics Institute, University College London, London, United Kingdom, 5 Wolfson Institute of Population Health, Barts and the London School of Medicine and Dentistry, Queen Mary University of London, London, United Kingdom, 6 Blizard Institute, Barts and the London School of Medicine and Dentistry, Queen Mary University of London, London, United Kingdom, 7 Manchester Centre for Genomic Medicine, Manchester University Hospitals NHS Foundation Trust, Manchester, United Kingdom, 8 Division of Evolution and Genomic Sciences, School of Biological Sciences, Faculty of Biology, Medicine and Health, University of Manchester, Manchester Academic Health Science Centre, Manchester, United Kingdom, 9 School of Basic and Medical Biosciences, Faculty of Life Sciences and Medicine, King's College London, London, United Kingdom, 10 Bradford Institute for Health Research, Bradford, United Kingdom, 11 British Heart Foundation Research Accelerator, University College London, London, United Kingdom, 12 Division of Psychiatry, University College London, London, United Kingdom, 13 London School of Hygiene & Tropical Medicine, London, United Kingdom

☉ These authors contributed equally to this work.
¶ Membership of Genes & Health Research Team is provided in the Acknowledgments.
* s.finer@qmul.ac.uk

**Data Availability Statement:** The GWAS summary statistics from Vujkovic et al. used to construct the

## Abstract

### Background

Type 2 diabetes (T2D) is highly prevalent in British South Asians, yet they are underrepresented in research. Genes & Health (G&H) is a large, population study of British Pakistanis and Bangladeshis (BPB) comprising genomic and routine health data. We assessed the extent to which genetic risk for T2D is shared between BPB and European populations (EUR). We then investigated whether the integration of a polygenic risk score (PRS) for T2D with an existing risk tool (QDiabetes) could improve prediction of incident disease and the characterisation of disease subtypes.

### Methods and findings

In this observational cohort study, we assessed whether common genetic loci associated with T2D in EUR individuals were replicated in 22,490 BPB individuals in G&H. We replicated fewer loci in G&H (*n* = 76/338, 22%) than would be expected given power if all EUR-

polygenic score are available through dbGaP under accession number phs001672.v4, and those generated in Genes & Health are available at https://www.genesandhealth.org/research/scientific-data-downloads. Genes & Heath imputed genotype data are available on EGA (ega-archive.org; study accession number: EGAS00001005373). Individual-level phenotypic data from Genes & Health will be made available to researchers upon application to Genes & Health, following their open access policy described https://www.genesandhealth.org/research/scientists-using-genes-health-scientific-research [genesandhealth.org].

**Funding:** Genes & Health and its research team (including SF, CG, DvH, HM, WN, RT, JW) have received core funding from Wellcome (WT102627, WT210561), the Medical Research Council (UK) (M009017), Higher Education Funding Council for England Catalyst, Barts Charity (845/1796), Health Data Research UK (for London substantive site), research delivery support from the NHS National Institute for Health Research Clinical Research Network (North Thames), and an Industrial Consortium supported by Takeda, Glaxo Smith Kline, Merck, Pfizer, NovoNordisk, Maze Pharmaceuticals, Bristol Myers Squibb. SH is funded by the NIHR for this research project (ACF-2018-26-002). SF was supported by a pump-priming grant from the Diabetes Research and Wellness Foundation (SCA/PP/12/19). HM and QH are funded by a Wellcome Trust Grant (206194) to the Wellcome Sanger Institute. NS is funded and RTL is part-funded by the BigData@Heart Consortium funded by the Innovative Medicines Initiative-2 Joint (Agreement number 116074). RT is supported by the National Institute for Health Research (NIHR) Biomedical Research Centre based at Guy's and St Thomas' NHS Foundation Trust and King's College London. The funders had no role in study design, data collection and analysis, decision to publish, or preparation of the manuscript.

**Competing interests:** I have read the journal's policy and the authors of this manuscript have the following competing interests: NS is now employed by GlaxoSmithKline.

**Abbreviations:** ANOVA, analysis of variance; AUC, area under the receiver operating characteristic curve; BMI, body mass index; BPB, British Pakistanis and Bangladeshis; CI, confidence interval; C-Index, concordance index; CUD, Clinically Undifferentiated Diabetes; EHR, electronic health record; FPG, fasting plasma glucose; GDM, gestational diabetes mellitus; G&H, Genes & Health; GWAS, genome-wide association study;

ascertained loci were transferable ($n = 101$, 30%; $p = 0.001$). Of the 27 transferable loci that were powered to interrogate this, only 9 showed evidence of shared causal variants. We constructed a T2D PRS and combined it with a clinical risk instrument (QDiabetes) in a novel, integrated risk tool (IRT) to assess risk of incident diabetes. To assess model performance, we compared categorical net reclassification index (NRI) versus QDiabetes alone. In 13,648 patients free from T2D followed up for 10 years, NRI was 3.2% for IRT versus QDiabetes (95% confidence interval (CI): 2.0% to 4.4%). IRT performed best in reclassification of individuals aged less than 40 years deemed low risk by QDiabetes alone (NRI 5.6%, 95% CI 3.6% to 7.6%), who tended to be free from comorbidities and slim. After adjustment for QDiabetes score, PRS was independently associated with progression to T2D after gestational diabetes (hazard ratio (HR) per SD of PRS 1.23, 95% CI 1.05 to 1.42, $p = 0.028$). Using cluster analysis of clinical features at diabetes diagnosis, we replicated previously reported disease subgroups, including Mild Age-Related, Mild Obesity-related, and Insulin-Resistant Diabetes, and showed that PRS distribution differs between subgroups ($p = 0.002$). Integrating PRS in this cluster analysis revealed a Probable Severe Insulin Deficient Diabetes (pSIDD) subgroup, despite the absence of clinical measures of insulin secretion or resistance. We also observed differences in rates of progression to micro- and macrovascular complications between subgroups after adjustment for confounders. Study limitations include the absence of an external replication cohort and the potential biases arising from missing or incorrect routine health data.

## Conclusions

Our analysis of the transferability of T2D loci between EUR and BPB indicates the need for larger, multiancestry studies to better characterise the genetic contribution to disease and its varied aetiology. We show that a T2D PRS optimised for this high-risk BPB population has potential clinical application in BPB, improving the identification of T2D risk (especially in the young) on top of an established clinical risk algorithm and aiding identification of subgroups at diagnosis, which may help future efforts to stratify care and treatment of the disease.

## Author summary

### Why was this study done?

- The common genetic changes associated with type 2 diabetes (T2D) have been extensively investigated in large studies of people from European ancestry. However, it is not known whether these findings can be transferred to people of South Asian origin, who are disproportionately affected yet underrepresented in genetic studies.

- Polygenic risk scores (PRSs) have emerged as a useful clinical tool with which to improve the prediction of who is/is not at risk of developing T2D, but they have not yet been assessed alongside existing predictive tools already used in routine clinical care, or to uncover "subtypes" of the condition.

HDL, high-density lipoprotein; HR, hazard ratio; IRD, Insulin-Resistant Diabetes; IRT, integrated risk tool; MARD, Mild Age-Related Diabetes; MD, Mild Diabetes; MOD, Mild Obesity-related Diabetes; NRI, net reclassification index; OR, odds ratio; PC, principal components; PRS, polygenic risk score; pSIDD, Probable Severe Insulin-Deficient Diabetes; SD, standard deviation; SIDD, Severe Insulin-Deficient Diabetes; SIRD, Severe Insulin-Resistant Diabetes; T2D, type 2 diabetes; UKBB, UK Biobank.

## What did the researchers do and find?

- We assessed whether the common genetic differences associated with T2D in people of European ancestry could be transferred to people of British Pakistani and Bangladeshi (BPB) ancestry ($n$ = 18,875). We found genetic differences between these ancestry groups that were significant.

- We built a T2D PRS for BPB ($n$ = 13,648) and integrated it with a clinical risk score (QDiabetes). Our integrated risk tool (IRT) improved the prediction of T2D, especially in individuals aged less than 40 years deemed low risk by QDiabetes alone. The PRS was also associated with the development of T2D after a pregnancy affected by gestational diabetes.

- Lastly, we used the PRS, in combination with standard clinical measures, to help elucidate subgroups of T2D in our study population ($n$ = 5,904) and differences in the risk of future diabetes complications.

## What do these findings mean?

- Our work highlights the need for greater representation of diverse ancestry groups in genetic studies of T2D.

- Integration of a PRS with clinical risk factors improved the prediction of T2D in BPB individuals, especially in the young.

- The T2D PRS can help to identify clinically distinct disease subgroups at diagnosis. Identification of these subgroups may support stratification of T2D care to improve health outcomes and allocate healthcare resources more efficiently in the future.

## Introduction

People of South Asian origin are disproportionately affected by type 2 diabetes (T2D) and tend to develop the condition at younger ages and at a lower body mass index than European ancestry individuals [1]. Despite this, they are underrepresented in studies assessing the genetic aetiology of the disease. Furthermore, to our knowledge, there has been no systematic assessment of the extent to which genetic risk loci identified in European ancestry individuals can be transferred into South Asians. Given the known pathophysiological differences between these populations [2,3], such an assessment is important for understanding the extent to which the aetiology of the disease varies between them.

Characterisation of the genetic aetiology of T2D, using genome-wide association studies, has allowed the development of polygenic risk scores (PRS) to aid the individualised clinical prediction of common complex diseases [4–10]. To date, PRS have been developed and tested predominantly in highly selected white European populations (EUR) with bias towards healthy and older people [10,11]. For coronary artery disease, integration of PRS with clinical risk tools has been shown to enhance the prediction of incident disease, in multiple ancestral groups [5,6,9,12], which may help target preventative care. Similarly, there is considerable potential to use PRS to improve the prediction of T2D risk [11], and, to date, their integration

with well-validated clinical risk instruments already in routine use, such as QDiabetes, has not yet been evaluated. Building enhanced risk tools that combine PRS with tools such as QDiabetes could offer significant opportunities for clinical benefit including enhanced individualised screening and preventive measures such as referral to diabetes prevention programmes [13]. There is a particular need to investigate PRS in understudied groups who are at high risk of T2D, including people of South Asian ancestry [11,14] and women with a history of gestational diabetes mellitus (GDM) [15]. PRS may also enhance the characterisation of T2D "subgroups," a recent area of significant research and clinical interest due to their potential to capture important heterogeneous features at diabetes diagnosis associated with common aetiological disease pathways that may predict future diabetes complications and treatment responsiveness [2,16–18].

In this study, we aimed to develop and evaluate a T2D PRS in British Bangladeshis and Pakistanis (BPBs) enrolled in the Genes & Health (G&H) programme [19]. This real-world, community-based cohort ($n$ = 48,144) combines genetic data with rich, lifelong electronic health record data and comprises a minority ethnic group living predominantly in socioeconomically deprived circumstances, otherwise underrepresented in health research [19]. We therefore aimed, firstly, to examine the transferability of T2D genetic loci already ascertained in individuals of European ancestry (EUR) to BPBs, taking into account power and differences in linkage disequilibrium, and to optimise a PRS for this population. Secondly, we tested whether the PRS enhanced the prediction of incident T2D when integrated with the commonly used clinical risk score, QDiabetes [20]. Thirdly, we sought to investigate whether the PRS alone might predict the progression to T2D from gestational diabetes in BPB women, as has been observed in European and Southeast Asian populations [21,22]. Finally, we explored whether the PRS might predict clinically heterogeneous T2D subgroups at diagnosis, which are increasingly well characterised [2,16–18,23].

## Methods

### Study population

G&H recruits BPB people aged 16 years and above, predominantly in community and primary care settings. We used the 2020 data release [24], which comprises electronic health record (EHR) data and genotype data from the Illumina Infinium Global Screening Array V3 Chip on 22,490 participants [19]. Descriptions of quality control and imputation of genotype data are provided in **S1 Text**. We applied specific inclusion and exclusion criteria to the G&H population for each analysis, described below and summarised in **Fig 1**. All analyses were planned at study outset, with the exception of the derivation of a second clustering model (Integrated Model), the rationale for which is given below.

### Ascertainment of conditions using clinical coding

T2D, GDM, and associated complications were ascertained using clinical codes extracted from primary and secondary care EHR, presented in full in **S1 Table**, developed using widely used clinical coding resources [25]. We excluded individuals with clinical codes of conditions causing secondary or monogenic diabetes.

### Transfer of previously identified GWAS loci to the study population

We obtained previously identified genetic loci associated with T2D from a genome-wide association study (GWAS) conducted in people of European ancestry (EUR) by Vujkovic and colleagues. [26]. We performed a GWAS of T2D in G&H and assessed if previously identified loci

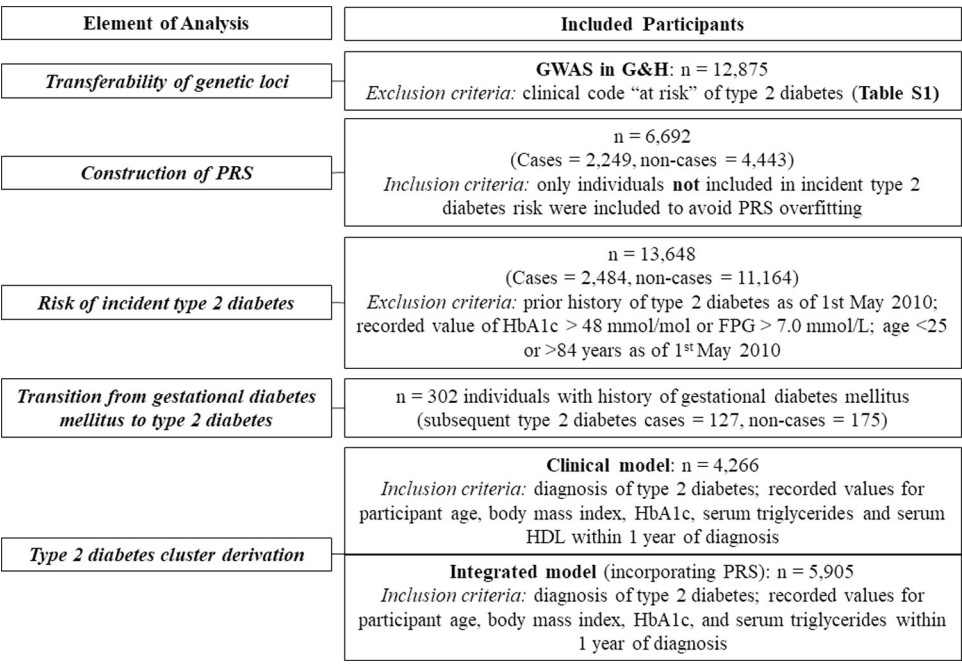

**Fig 1. Summary of analyses in study.** FPG, fasting plasma glucose; GWAS, genome-wide association study; G&H, Genes & Health; HDL, high-density lipoprotein; PRS, polygenic risk score.

were reproducible in G&H at *p*-value <0.05 using previously established methods (**S1 Text**) [9]. We assessed 338 T2D loci that had variants well imputed in G&H. The expected power for replication was estimated assuming the same effect size as in the EUR discovery sample and accounting for the allele frequency and sample size in G&H. Even though the same locus and target gene may affect T2D risk in both populations, it is possible that this is caused by distinct causal variants in the region. Therefore, trans-ancestry colocalisation analysis was performed using TEColoc to assess whether a transferable locus shared the same causal variant between BPB and EUR populations, using UK Biobank (UKBB) EUR individuals for the latter [27]. We used overlapping variants in a 50-kb window and assessed 27 transferable loci in which ≥10% of the variants from UKBB were well imputed in G&H and vice versa. We might not be able to detect shared causal variants due to the low coverage for some loci, but it is not likely to cause false positives. We also applied the Popcorn algorithm to estimate the trans-ancestry genetic correlation between BPB and UKBB EUR populations [28]. Further methodological details are given in **S1 Text**.

## Polygenic risk score construction

We applied previously constructed scores from the PGS Catalog [29], principally developed in EUR populations, to participants in G&H. These were compared to PRS optimised within G&H using the clumping and *p*-value thresholding (C+T) method implemented in PRSice2 v2.2.11 [30,31] based on the largest published T2D multiancestry GWAS to date [26] (**S1 Fig**), downloaded from dbGaP (phs001672.v4). While the EUR-specific GWAS was used for assessing the transferability of loci as mentioned above, we constructed the PRS using the more powerful multiancestry GWAS, which contained 9,004 cases and 12,066 controls with Pakistani ancestry. We also tried using a South Asian–specific GWAS [32] to construct a PRS and combining the PRS from the South Asian–specific GWAS with the one from the multiancestry

GWAS using the method from Marquez-Luna and colleagues [33] (**S1 Text**). Neither of the scores showed better performance; thus, we used the C+T PRS constructed from the multiancestry GWAS. For the C+T score, LD estimated from EUR samples ($N$ = 503) from the 1,000 Genomes project was used for clumping ($r^2$ = 0.1). We calculated multiple scores using various *p*-value thresholds. We excluded one sample from each pair of 2nd degree relatives identified using the kinship coefficient from KING v2.3.4 [34] and assessed the accuracy of PRS in unrelated individuals. We selected the PRS with the highest predictive accuracy in an independent set of samples that were not included in the QDiabetes analysis described below, i.e., prevalent cases with onset before the assessment date, and controls who did not meet inclusion criteria for longitudinal analysis (aged 25 to 84 years with no prior history of T2D as of 1 May 2010, no recorded value of HbA1c >48 mmol/mol or FPG >7.0 mmol/L; see **S2 Table** for characteristics of cases and non-cases). The area under the receiver operating characteristic curve (AUC) was estimated with the R package "pROC." Predictive accuracy of PRS was quantified by incremental AUC, or the gain in AUC when adding PRS to the reference model, which accounted for participant age, gender, and 10 genetic principal components (PCs). We also calculated the incremental pseudo $R^2$ on the liability scale [35]. We used 13.6% as the prevalence estimate for T2D in South Asian ancestry individuals in the UK, the background population from which G&H is sampled, defined as all people from South Asian ethnicities ($n$ = 255,066 aged ≥20 years) registered with a primary health physician/GP in 4 east London boroughs. The C+T PRS with *p*-value threshold $< 1 \times 10^{-5}$ (2,877 SNPs; **S3 Table**) showed the highest predictive accuracy among all the scores we constructed and was thus used in downstream analysis (**S1 Text**; **S1 Fig**). A scaled PRS following the normal distribution with mean and median of 0 and standard deviation (SD) of 1 was constructed across BPB participants after regressing out the first 10 PCs derived as described in **S1 Text**, allowing direct comparison between ancestry groups [36]. We used additional PRSs associated with T2D and its aetiology that are previously described in the literature by Mansour Aly and colleagues [18] and Udler and colleagues [37] (**S1 Text**).

## Development of an integrated risk tool to predict incident type 2 diabetes

We assessed whether PRS could enhance 10-year risk prediction of T2D compared to QDiabetes, a validated, EHR-based risk prediction tool commonly used in UK primary care to estimate an individual's 10-year risk of developing T2D [20]. There are 3 QDiabetes models. Model A provides estimates based on age, ethnicity, family history, comorbidities, and prescribed medications. Model B (which has the highest clinical predictive value) uses the same variables as model A, plus fasting plasma glucose (FPG), while model C is composed of model A plus HbA1c. Analysis was performed in 13,648 individuals aged 25 to 84 years with no prior history of T2D as of 1 May 2010, no recorded value of HbA1c >48 mmol/mol or FPG >7.0 mmol/L (**S1 Table**), using established inclusion criteria [20]. QDiabetes scores were calculated on the assessment date of 1 May 2010 for each participant on the basis of available clinical data required to run the model using the R package "QDiabetes" [38]. Numbers available for analysis were model A ($n$ = 13,648), model B ($n$ = 4,334), and model C ($n$ = 864). We applied multiple imputation to fill in missing data for body mass index (BMI) and the Townsend Deprivation Index using the R package "MICE" [39].

We combined the PRS and QDiabetes score, which were not significantly correlated, to build an integrated risk tool (IRT) to estimate risk of developing T2D over the next 10 years, as previously described [12]. For each individual, QDiabetes score was converted to an odds ratio (OR) and multiplied by the odds calculated from the individual's PRS given their QDiabetes score, calculated from a logistic regression model incorporating PRS, QDiabetes, and an

interaction between them, and training it separately on male and female participants. Ten-year risk of disease was classified as high (>10%) or low (<10%) in line with [12]. We assessed relative model performance using the concordance index (C-Index) and categorical net reclassification index (NRI). C-Indices were calculated from Cox proportional hazard models using the R package "survival." NRI was calculated as NRI = P(up|case) − P(down|case) + P(down|noncase) − P(up|noncase), i.e., the sum of (1) the proportion of individuals who subsequently developed T2D correctly reclassified as high risk, minus the proportion of individuals who subsequently developed T2D incorrectly reclassified as low risk; and (2) the proportion of individuals who did not develop T2D correctly reclassified as low risk, minus the proportion of individuals who did not develop T2D incorrectly reclassified as high risk. Categorical NRI performance was assessed in the entire analysed sample, plus in age-by-sex subgroups, with 40 years chosen as the threshold for high/low age group comparison to reflect (1) a mean age at T2D diagnosis of 49.1 years and median of 47.8 years in our cohort; (2) a mean age at study entry of 39.1 years; (3) the fact that routine NHS health checks that include T2D screening are offered at age 40; and (4) previous use of age 40 as a cutoff discriminating early from late-onset diabetes in the literature [40]. We observed that findings were robust to alteration of this threshold across a broad age range (**S2 Fig**). NRI confidence intervals were calculated using bootstrapping (number of iterations = 1,000). The characteristics of reclassified individuals were compared with descriptive statistics. In addition to the categorical NRI, at the request of a peer reviewer, we also calculated continuous NRI using the R package "PredictABEL." P(up|case) and P(down|case) in the above equation now indicate the proportion of cases that had higher or lower risk estimates using the IRT than QDiabetes, respectively.

## Prediction of type 2 diabetes after gestational diabetes using PRS

In women with a history of GDM, we compared characteristics (including T2D PRS) of those who subsequently did ($n$ = 127) and did not ($n$ = 175) develop T2D. We compared characteristics between the 2 groups using multivariate logistic regression models that included risk factors used to construct the QDiabetes score. Details of a power calculation for this analysis are available in **S1 Text**. The association between PRS and T2D in women with a history of GDM was assessed in Cox proportional hazard models controlling for (1) QDiabetes score or (2) unadjusted clinical risk factors used to construct the QDiabetes score.

## Identification of type 2 diabetes subgroups and their association with PRS

We applied data-driven clustering techniques to define subgroups of individuals with distinct characteristics in 2 separate models using latent class analysis (Stata V16.0) based on the methodology used by [2,16,17]. The optimal number of clusters was selected on the basis of information criteria elbow plots (**S3 Fig**) and was found to be 5 for all model iterations. The first model (Clinical Model) was based on 5 clinical variables at the time of T2D diagnosis: age, BMI, HbA1c, serum triglycerides, and serum high-density lipoprotein (HDL). Analysis was restricted to 4,266 participants for whom values for these variables were available within 1 year of T2D diagnosis. Differences in PRS between clusters were compared using analysis of variance (ANOVA) (RStudio V1.1.413). Having observed differing PRS distribution between clusters derived using clinical variables, we next explored whether inclusion of PRS as a clustering covariate could further delineate the aetiological processes underlying cluster membership. In the second model (Integrated Model), we repeated clustering using age, BMI, HbA1c, serum triglycerides, and PRS as covariates in 5,905 individuals with available data within 12 months of diagnosis. HDL was omitted from the Integrated Model as its inclusion did not alter cluster characteristics but did reduce sample size. Differences in previously reported glycaemic trait

PRSs including fasting glucose, beta cell function, insulin sensitivity index, and corrected insulin response were compared between each cluster and nondiabetic controls ($n$ = 10,841) using Bonferroni-corrected unpaired one-tailed $t$ tests [18,37]. Further details are presented in **S1 Text**.

### Research ethics and reporting guidelines

G&H operates under approval from the National Research Ethics Committee (London and Southeast), and the Health Research Authority (reference 13/LO/124); Queen Mary University of London is the Sponsor. Written informed consent is obtained from all study volunteers, and it allows analysis of health and genetic data and publication of results. This study is reported according to the Transparent reporting of a multivariable prediction model for individual prognosis or diagnosis (TRIPOD) statement (checklist available in **S2 Text**).

## Results

As of 1 May 2020 (the end of the study period), 7,599 individuals with an EHR diagnosis of T2D were enrolled in G&H, followed up for a mean of 9.69 years after diagnosis. 52.8% of these individuals were in the most deprived Index of Multiple Deprivation quintile in the UK. Mean age at time of diagnosis was 46.2 years. A total of 1,205 individuals (15.9%) developed macrovascular complications between T2D diagnosis and the end of the study period; 2,300 (30.3%) developed microvascular complications.

### Genetics of type 2 diabetes in BPB versus European individuals

We first investigated the extent to which genetic risk for T2D was shared between G&H (BPB) and EUR individuals. The GWAS in G&H identified 3 significant associations at $p$-value $< 5 \times 10^{-8}$ (**S4 Table**, **S4 Fig**), and they were all previously identified [26]. Four out of the 6 genetic loci identified in the South Asian–specific GWAS by Kooner and colleagues [32] were replicated in G&H (**S5 Table**). The trans-ancestry genetic correlation (i.e., the correlation of causal-variant effect sizes) between G&H and UKBB EUR individuals was significantly lower than 1 ($r_g$ = 0.68, standard error = 0.15, $p$-value (for the null hypothesis that $r_g$ = 1) = 0.03). Among the 338 genetic loci identified in EUR populations that had variants well imputed in G&H, we observed significant evidence of transferability for 76 (22.5%) loci (among them 13 were significant at $p$-value $< 0.05/338$; **S6 Table**), which was lower than would be expected (30.0%) accounting for differences in power (one-sided binomial $p$-value = 0.001). This suggested that a large proportion (75%) of the loci ascertained in EUR populations, which were well powered to replicate in G&H did so. We did not observe any loci that were well powered (power to replicate >0.8) but not transferable in G&H. The evidence of transferability was consistent with other cardiometabolic traits (Observed/Expected = 0.62 for coronary artery disease and 1.0 to 1.2 for BMI, lipids, and blood pressure) that were reported in the same cohort [9].

To assess whether causal variants were shared between EUR and BPB populations for the transferable loci, we applied trans-ancestry colocalisation analysis. Of 27 replicated loci assessed, 9 (33%) had significant evidence of shared causal variants (**S7 Table**). For example, we observed shared causal variants at the *TCF7L2* locus, one of the known loci with the strongest association with T2D, and at the *KCNJ11* locus, which is the target gene for drugs such as Glyburide [41,42]. The proportion of transferable loci with shared causal variants for T2D was lower than for triglycerides (56%; binomial $p$-value = 0.015) and total cholesterol (61%; $p$ = 0.003) and similar to HDL (48%), LDL (47%), and BMI (26%) (binomial $p$-value >0.05 for all) in the same cohort [9].

**Table 1. Characteristics of participants included in analysis with each QDiabetes model.**

| | Model A (N = 13,648) | Model B (N = 4,334) | Model C (N = 864) | p-value |
|---|---|---|---|---|
| % Developing T2D | 18.2 | 27.0 | 33.3 | <0.001 |
| T2D PRS | −0.046 (0.99) | −0.080 (0.99) | −0.062 (1.01) | 0.210 |
| Age (Years) | 35.4 (8.8) | 42.0 (11.1) | 45.2 (12.3) | <0.001 |
| % Female | 49.8 | 59.2 | 64.9 | <0.001 |
| % Bangladeshi | 69.9 | 73.9 | 58.0 | <0.001 |
| % Family history of diabetes | 35.9 | 40.2 | 33.9 | <0.001 |
| Index of Multiple Deprivation (2015) score | 7.4 (2.2) | 7.8 (1.9) | 7.2 (2.4) | <0.001 |
| BMI (kg/m²) | 25.8 (4.6) | 26.704 (4.624) | 27.7 (4.7) | <0.001 |
| HbA1c (mmol/mol) | 39.1 (4.3) | NA | 38.0 (4.2) | 0.271 |
| Fasting glucose (mmol/L) | 4.9 (0.7) | 4.8 (0.7) | NA | 0.154 |
| HDL (mmol/L) | 1.2 (0.3) | 1.2 (0.3) | 1.2 (0.3) | <0.001 |
| Triglycerides (mmol/L) | 1.9 (1.3) | 1.8 (1.2) | 1.7 (1.2) | 0.077 |
| **Preexisting Conditions** | | | | |
| Gestational diabetes (% females) | 4.1 | 5.2 | 10.7 | <0.001 |
| Polycystic ovarian syndrome (% females) | 4.7 | 7.3 | 6.6 | <0.001 |
| Cardiovascular disease (%) | 2.5 | 4.7 | 9.7 | <0.001 |
| Hypertension (%) | 6.1 | 14.2 | 13.7 | <0.001 |

QDiabetes Model A is calculated in individuals without HbA1c or FPG; Model B by adding FPG to Model A; and Model C by adding HbA1c to Model A. Values show the mean with SD in brackets, unless otherwise indicated. Means were compared with ANOVA.

ANOVA, analysis of variance; BMI, body mass index; FPG, fasting plasma glucose; HbA1c, hemoglobin A1c; HDL, high-density lipoprotein; PRS, polygenic risk score; SD, standard deviation; T2D, type 2 diabetes.

## Performance of an integrated risk tool to predict incident type 2 diabetes

The characteristics of participants included in incident T2D risk prediction analyses are shown in **Table 1**. We constructed a PRS for T2D using the clumping and $p$-value thresholding method, based on multiancestry GWAS meta-analysis [26]. This had an OR per SD of 1.57 (95% confidence interval [CI]: 1.50 to 1.65), an incremental AUC of 0.032 (95% CI: 0.026 to 0.039), and an incremental $R^2$ on the liability scale of 4.6% (95% CI: 3.7 to 5.6) in G&H. There was no correlation between PRS and QDiabetes scores (Pearson's coefficients −0.03, 0.08, 0.13 for QDiabetes models A, B, and C, respectively; associated $p$-values 0.31, 0.18, 0.16) (**S8 Table**). PRS was weakly correlated with fasting glucose (Pearson's coefficient 0.11, $p < 0.001$) and HbA1c (Pearson's coefficient 0.07, $p = 0.048$) (**S8 Table**). Compared to QDiabetes alone, the IRT combining QDiabetes model A (multiple clinical risk factors) with PRS improved the 10-year prediction of T2D as assessed by categorical NRI: NRI 3.22% (95% CI: 2.00% to 4.38%) in 13,648 individuals (**Fig 2A**). The IRT C-index, a goodness-of-fit metric approximating the area under the receiver–operator curve, was superior to QDiabetes model A for participants aged less than 40 years ($p = 0.002$) (**Fig 3A, S9 Table**), but not for people aged over 40 years. Enhancement of T2D prediction with the IRT persisted but was attenuated when QDiabetes model B (model A plus FPG) was used: categorical NRI 0.80% (95% CI 0.21% to 1.42%). The IRT did not improve on the QDiabetes score with model C (model A plus HbA1c): categorical NRI 0.20% (95% CI: −0.09% to 0.44%) (**Fig 2B and 2C, S9 Table**). Higher estimates of model enhancement were observed with continuous NRI (**S9 Table**): Model A 28.1% (95% CI 23.9% to 32.1%; Model B 23.8% (95% CI 17.2% to 30.6%); Model C 19.4% (95% CI 4.92% to 33.9%). There were no observed differences in C-Index between QDiabetes and IRT in models B and C (**Fig 3B and 3C, S9 Table**).

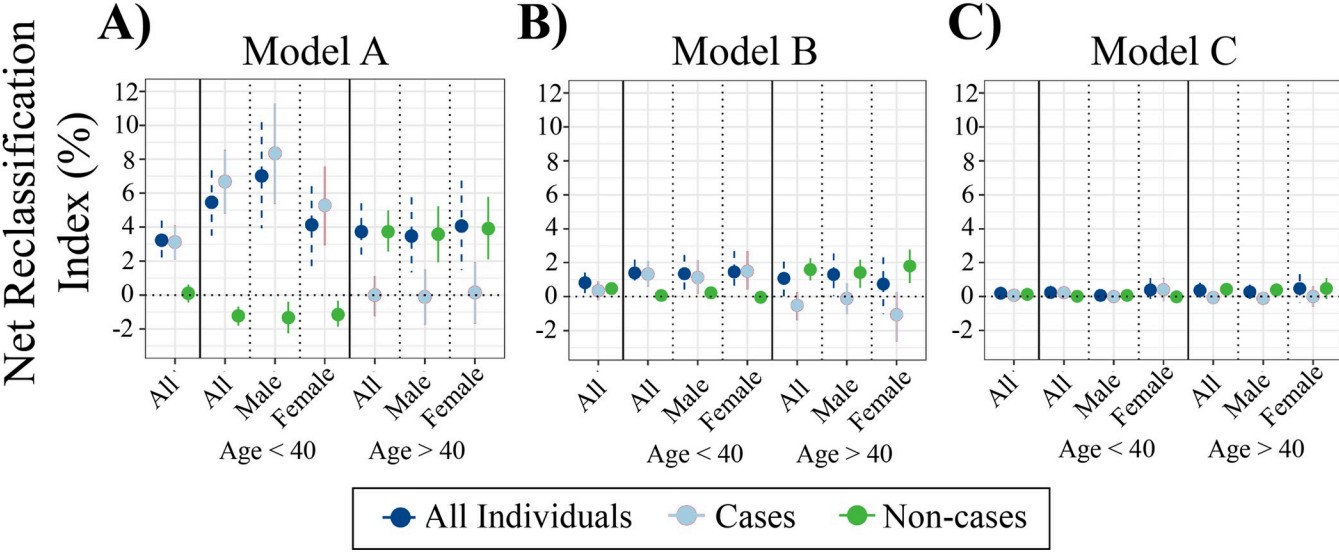

**Fig 2. Performance of the IRT using NRI.** NRI and 95% CIs comparing 10-year prediction of T2D between QDiabetes models A ($n$ = 13,648), B ($n$ = 4,334), and C ($n$ = 864) and IRTs combining each QDiabetes model with T2D PRS. NRI is presented overall and for age-by-sex subgroups. The NRI for individuals who subsequently develop T2D (cases) and who do not develop T2D (non-cases) are presented alongside the NRI for cases and non-cases combined (all individuals). CI, confidence interval; IRT, integrated risk tool; NRI, net reclassification index; PRS, polygenic risk score; T2D, type 2 diabetes.

Across all IRT models (for QDiabetes model A, B, and C), NRI was higher in participants aged less than 40 years who subsequently developed T2D (cases) than those who did not (non-cases), implying improved ability to correctly classify younger individuals as high risk. The converse pattern was seen in individuals aged greater than 40 years, implying enhanced ability to correctly classify older individuals as low risk. The clinical features of individuals whose risk

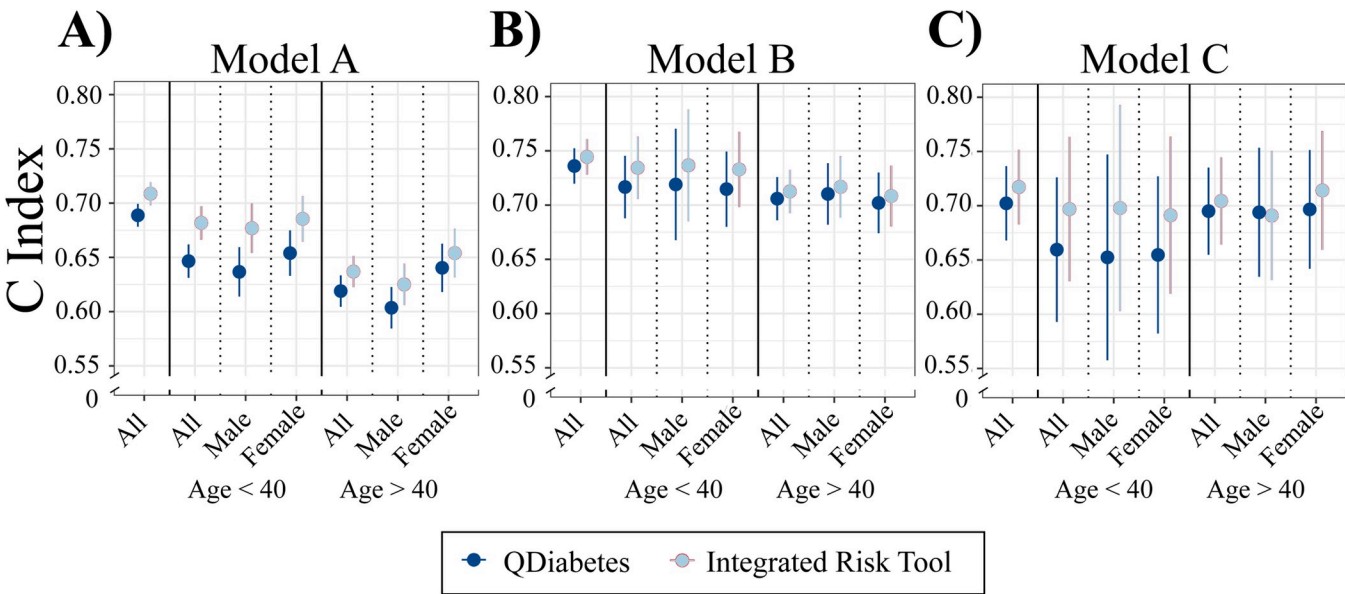

**Fig 3. Performance of the IRT using C-Index.** Comparison of C-Index calculated from Cox proportional hazard models for QDiabetes models A, B, and C and IRTs combining each QDiabetes model with individual T2D PRS. C-index is presented for all participants and for age-by-sex subgroups. Data are presented as mean with 95% CIs. CI, confidence interval; C-Index, concordance index; IRT, integrated risk tool; PRS, polygenic risk score; T2D, type 2 diabetes.

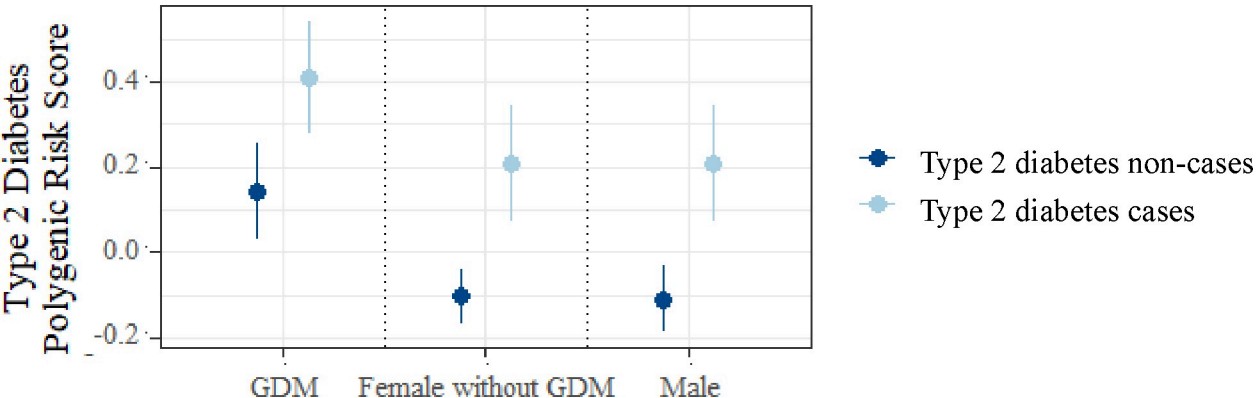

**Fig 4. Mean T2D PRS in women with GDM.** Mean T2D PRS in women with GDM who subsequently did and did not develop T2D, compared to age- and BMI-matched control groups of females without GDM, and males. Data are presented as group mean with 95% CIs according to sex-by-GDM subgroup, stratified by subsequent T2D status. BMI, body mass index; CI, confidence interval; GDM, gestational diabetes mellitus; PRS, polygenic risk score; T2D, type 2 diabetes.

was reclassified are presented in **S10 Table.** The number of individuals reclassified in each model are presented by age band and sex in **S11 Table.**

### Prediction of type 2 diabetes after gestational diabetes using PRS

In women who had a history of GDM, T2D PRS was higher in those who had subsequently developed T2D, compared to those who had not (mean value (SD) 0.408 (0.93) versus 0.140 (0.98); $p = 0.02$) (**Fig 4**). Among individuals without T2D (non-cases), women with a history of GDM displayed a higher mean PRS than age- and BMI-matched females without GDM and compared to males ($p = 0.003$ and $p = 0.001$, respectively) (**Fig 4**). In multivariate survival analysis restricted to women with a history of GDM, T2D PRS was independently associated with development of T2D after adjustment for (1) QDiabetes score model A (hazard ratio (HR) per SD of PRS 1.23, 95% CI 1.05 to 1.42; $p = 0.028$) and (2) established risk factors for T2D (HR 1.37, 95% CI 1.17 to 1.57; $p = 0.002$) (**S12 Table**). Similar findings were observed in age- and BMI-matched male and female (without a history of GDM) controls (**S12 Table**), indicating that the utility of the T2D PRS for risk prediction does not appear to differ between women with versus without GDM.

### Type 2 diabetes subgroup identification using clustering

In our Clinical Model, we identified T2D subgroups by clustering age, BMI, HbA1c, HDL, and triglycerides at the time of diagnosis (**Fig 5A, Table 2A**). We replicated some previously identified clusters: Mild Obesity-related Diabetes (MOD), Mild Age-Related Diabetes (MARD), Severe Insulin-Resistant Diabetes (SIRD), and Mild Diabetes (MD), and used their previous nomenclature [2,16,17,43]. In the absence of clinical measures of insulin secretion such as C-peptide, we were unable to fully delineate the previously well-replicated Severe Insulin-Deficient Diabetes (SIDD) cluster in the Clinical Model. However, our MD cluster may include individuals who are insulin deficient as it is characterised by low serum triglycerides (1.21 (0.43) mmol/L) in addition to high HDL (2.07 (0.25) mmol/L), and our Clinically Undifferentiated Diabetes (CUD) cluster likely also contains people with SIDD as its membership is characterised by high HbA1c and low serum triglycerides. There were no statistically significant differences in rates of progression to micro- or macrovascular complications between clusters identified in the Clinical Model. However, we found that the PRS differed significantly

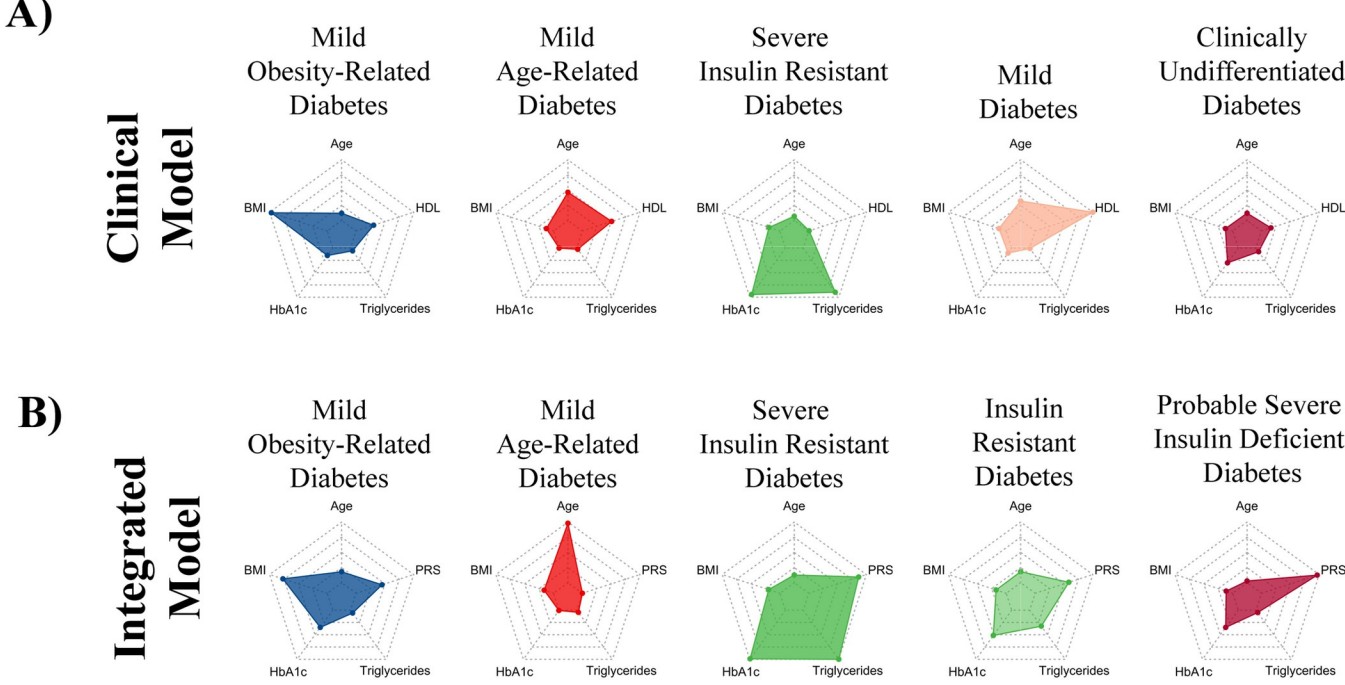

**Fig 5. Cluster analysis using clinical variables at T2D diagnosis with and without PRS.** Radar plots showing cluster mean values of the variables used to derive 5 clusters of individuals with T2D at the time of diagnosis in 2 separate models. In panel A (Clinical Model), latent class analysis was performed using age, BMI, HbA1c, serum triglycerides, and HDL. In panel B (integrated model), clustering was repeated using T2D PRS in place of HDL. The centre and edge of each polygon represent the minimum and maximum mean cluster values for each variable, respectively. All polygons represent the same scale; all scales are linear. Mean values for each variable within each cluster are represented by coloured dots. For example, mean BMI is highest in cluster 1. BMI, body mass index; HbA1c, hemoglobin A1c; HDL, high-density lipoprotein; PRS, polygenic risk score; T2D, type 2 diabetes.

**Table 2. Clinical and Integrated cluster analyses.**

| A<br>Clinical Model | | MOD (N = 211) | MARD (N = 861) | SIRD (N = 32) | MD (N = 95) | CUD (N = 3,067) | *p*-value |
|---|---|---|---|---|---|---|---|
| | **Age (Years)** | 44.5 (11.4) | 50.9 (12.2) | 43.1 (10.9) | 49.3 (12.3) | 45.2 (10.6) | <0.001 |
| | **BMI (kg/m²)** | 40.2 (3.6) | 27.6 (4.0) | 28.2 (3.9) | 27.1 (5.2) | 27.5 (3.7) | <0.001 |
| | **HbA1c (mmol/mol)** | 57.5 (15.8) | 53.4 (13.4) | 71.9 (26.9) | 55.9 (17.63) | 59.0 (16.4) | <0.001 |
| | **Serum triglycerides (mmol/L)** | 1.72 (0.73) | 1.39 (0.63) | 12.2 (3.95) | 1.21 (0.43) | 1.97 (1.08) | <0.001 |
| | **HDL (mmol/L)** | 1.17 (0.24) | 1.45 (0.15) | 0.79 (0.21) | 2.07 (0.20) | 0.99 (0.17) | <0.001 |
| | **PRS** | 0.07 (1.02) | 0.23 (0.97) | 0.29 (1.14) | 0.26 (1.09) | 0.34 (0.97) | 0.002 |
| B<br>Integrated Model | | MOD (N = 556) | MARD (N = 1180) | SIRD (N = 37) | IRD (N = 579) | pSIDD (N = 3,489) | *p*-value |
| | **Age (Years)** | 44.8 (9.6) | 61.6 (7.6) | 43.7 (11.3) | 42.8 (9.9) | 41.6 (7.6) | <0.001 |
| | **BMI (kg/m²)** | 37.2 (3.9) | 27.5 (3.4) | 28.1 (3.7) | 27.8 (3.8) | 26.7 (3.3) | <0.001 |
| | **HbA1c (mmol/mol)** | 60.0 (17.1) | 53.1 (11.6) | 72.8 (26.8) | 63.3 (19.1) | 60.0 (16.5) | <0.001 |
| | **Serum triglycerides (mmol/L)** | 1.77 (0.73) | 1.62 (0.70) | 13.0 (3.63) | 4.97 (1.05) | 1.66 (0.74) | <0.001 |
| | **PRS** | 0.16 (0.91) | −0.10 (0.92) | 0.41 (0.96) | 0.25 (1.02) | 0.48 (0.95) | <0.001 |

Distribution of clustering variables and T2D PRS across 5 data-driven clusters of individuals with T2D at the time of diagnosis in a clinical (panel A) and integrated clinical and genetic (panel B) model. PRS was not used as a clustering variable in the clinical model; it was used as a clustering variable in the integrated model. In the clinical model, the distribution of PRS between groups was compared after cluster allocation. Data are presented as mean (standard deviation). Mean values were compared using ANOVA.

ANOVA, analysis of variance; BMI, body mass index; CUD, Clinically Undifferentiated Diabetes; HbA1c, hemoglobin A1c; HDL, high-density lipoprotein; IRD, Insulin-Resistant Diabetes; MARD, Mild Age-Related Diabetes; MD, Mild Diabetes; MOD, Mild Obesity-related Diabetes; PRS, polygenic risk score; pSIDD, Probable Severe Insulin-Deficient Diabetes; SIRD, Severe Insulin-Resistant Diabetes.

between clusters (ANOVA, $p = 0.002$) and that CUD was characterised by the highest observed PRS (0.34 (0.97)). PRS was lowest in MOD and MARD, with mean PRS being 0.07 (1.02) and 0.23 (0.97), respectively.

Given these results, we next explored whether inclusion of PRS as a covariate in the clustering model could further delineate the subgroups by revealing their likely aetiology (henceforth termed "Integrated Model"). We included PRS in the latent class analysis along with age, BMI, HbA1c, and triglycerides (**Fig 5B, Table 2B**). We identified broadly similar clusters, although with some additional differentiation not seen in the Clinical Model: MOD and MARD were still identified, but in the latter, mean age was notably higher than in the Clinical Model (61.6 (7.6) versus 50.9 (12.2) years). In this analysis, 2 clusters representing insulin-resistant diabetes were identified, differentiated by both the severity of hypertriglyceridaemia and PRS, and therefore, we call these SIRD and Insulin-Resistant Diabetes (IRD). The MD cluster was not identified in this analysis. Instead, this integrated analysis identified a cluster associated with a high PRS (mean PRS 0.48 (0.95)), young age of onset (41.6 (7.6) years), low BMI (26.7 (3.3) kg/m$^2$), and low triglycerides (1.66 (0.74) mmol/L). Given that our PRS is constructed predominantly with variants that are known to have strong effect sizes on T2D (**S7 Table**) and that these loci are well known [44] to represent defects in beta-cell function (including *TCF7L2*, *KCNJ11*, *HNF4A*, *CDKAL1*, *MTNR1B*, *SLC30A8*, and *IGF2BP2*), we call this cluster Probable Severe Insulin-Deficient Diabetes (pSIDD). Mean differences were compared between nondiabetic controls and each cluster for previously identified glycaemic trait PRSs [18,37] (**S5 Fig, S13 Table**). Compared to nondiabetic controls, the pSIDD cluster had higher fasting glucose PRS (mean difference 0.298, one-sided $p < 0.001$) and the beta cell PRS (mean difference 0.302, one-sided $p < 0.001$). Additionally, we saw increasing odds of allocation to pSIDD with increasing quintiles of fasting glucose PRS (OR for top versus bottom PRS quintile 1.15 (95% CI 1.09 to 1.20) and beta cell PRS (OR for top versus bottom PRS quintile 1.16 (95% CI 1.10 to 1.21)), and decreasing odds of allocation to pSIDD with increasing corrected insulin response PRS (OR for top versus bottom PRS quintile 0.87 (95% CI 0.83 to 0.91). These associations of pSIDD with glycaemic traits are in keeping with previous reports of SIDD in the literature [18]. The beta cell PRS was also significantly higher in SIRD (mean difference 0.339, one-sided $p = 0.02$) and IRD (mean difference 0.239, one-sided $p < 0.001$) than in controls, suggesting the possibility of combined insulin resistant and deficient aetiology underlying these clusters, as described in other studies of South Asians [2].

In Cox proportional hazard models (**S6 Fig**), macrovascular complication rates were higher in MARD and MOD ($p < 0.001$ and $p = 0.009$, respectively); microvascular rates were highest in IRD and MARD ($p < 0.001$ and $p = 0.002$). In age-adjusted Cox proportional hazard models, rates of progression to micro- and macrovascular complications in the MARD cluster were not significantly different to other clusters, but results for MOD and IRD were unchanged.

## Discussion

In this study, we harness the power of a large population-based study of BPB people with linked health and genomic data to improve the prediction of T2D. For the first time in a real-world study population, we show that use of PRS enhances prediction of incident T2D on top of an established and clinically validated risk prediction tool, QDiabetes. We also show that among BPB women with a history of GDM, PRS is associated with T2D development. Additionally, we show that our PRS is variably associated with T2D subgroups and can itself distinguish a subgroup that is undifferentiated by clinical features.

To our knowledge, this is the first study to systematically assess the transferability of genetic loci associated with T2D in EUR individuals into BPB individuals. Previous studies have

assessed directional consistency of genetic effects [45] or heterogeneity in allelic ORs [46] across ancestry groups but, to our knowledge, have not explicitly asked how many genetic loci one might expect to replicate in a sample of a given ancestry given the power and linkage disequilibrium differences. We replicated a lower percentage of previously identified GWAS loci (22.5%) in G&H than would be expected (30.0%) after accounting for power, consistent with the only moderate trans-ancestry genetic correlation between the 2 populations. That fewer loci were replicated than expected might also be due to ancestry-specific gene–environment interactions and the Winner's Curse phenomenon, an ascertainment bias in which the effect sizes of previously reported loci were likely overestimated, which could lower the chances of replication [47]. Among the genomic loci, which did transfer between EUR and BPB individuals and were powered to interrogate this, only 33% showed evidence of shared causal variants. Hence, despite the considerable overlap in genetic risk between BPB and EUR individuals, our results should still motivate larger studies of T2D genetics in different ancestry groups [48] in order to better characterise the genetic contribution to disease and ancestry-specific aetiological features. In the future, the findings of such studies may point to different effects of risk factors via mendelian randomisation, or different potential drug targets across populations.

Our IRT that combines QDiabetes scores with PRS modestly improved risk classification in a population at high risk of T2D. Specifically, the IRT improved NRI compared to QDiabetes models A (3.22%) and B (0.80%). The IRT did not improve NRI compared to model C, which may reflect reduced power in model C due to lower sample size ($n = 864$) than the other 2 models ($n = 13,648$ for model A; $n = 4,344$ for model B). Alternatively, the lower NRI for models B and C could imply that the benefit of integrating genetic information with risk models already incorporating metrics of hyperglycaemia is more limited than models without hyperglycaemic metrics (Model A). This hypothesis is supported by the association of HbA1c and fasting glucose with PRS (correlation coefficients 0.11 and 0.07, respectively), although the strength of the association is weak. This interpretation would be in line with previous findings reported in EUR individuals by Lyssenko and colleagues [49] and Meigs and colleagues [50], although as these studies were performed prior to the availability of any large GWAS data reassessment with an updated PRS is of benefit, as is its reporting in BPB individuals. Hippisley-Cox and Coupland [20] describe model B (incorporating FPG) as the best-fitting model in 11.5 million (multiethnic, but dominantly Europeans with only approximately 4.5% of South Asian ancestry) individuals in the QResearch database (C-Index 0.89 for women, 0.87 for men for model B, compared to 0.83 for women and 0.81 for men in model A), but we observed lower C-Index values for QDiabetes alone in our study (**Fig 3**). The lower C-Index values for comparable models in our study could be also due to differing age, and deprivation distributions between the QResearch database and G&H participants, and the lack of ethnic diversity in our model, since ethnicity is strongly weighted in the QDiabetes risk estimation algorithms and explains a large proportion of variance in T2D risk.

We observed positive NRIs with the IRT driven principally by enhanced reclassification of younger individuals from low to high risk and improved downgrading of older individuals from high to low risk. In the context of a growing burden of T2D in increasingly resource-constrained health systems, such reclassification would support more effective resource allocation, with intensification of preventative care pathways for those at highest risk (e.g., early referral to T2D prevention programmes [13]) and relaxation of those who are identified to be at reduced risk. We found that individuals being reclassified as higher risk tended to be young (mean age 36.0 years), free from comorbidities, and relatively slim (mean BMI 25.9 kg/m$^2$; **Table 2**). According to estimates by the International Diabetes Federation, half of cases of T2D remain undiagnosed [51]. Using our IRT to identify these individuals, who would likely otherwise have been considered healthy, as high risk could offer significant opportunities to identify

and manage T2D early and prevent subsequent morbidity and mortality [52]. This finding is particularly important given the observation that early-onset T2D is associated with rapid progression to vascular complications [53].

We showed that in BPB women with a history of GDM, a T2D PRS was associated with development of T2D. This finding persisted after adjustment for established clinical risk factors and the QDiabetes risk score. Our findings are in keeping with prior reports of an association between a T2D PRS and development of T2D in white [21] and Southeast Asian [22] women with a history of GDM. While previous studies have applied alternative T2D PRSs to women of South Asian origin to identify those at higher risk of GDM [54], none to date have explored their association with T2D development in women with GDM. Although we observed similar associations between PRS and T2D development in age- and BMI-matched male and female controls, the clinical utility of this finding in women with a history of GDM is clearer: this patient group is at extremely high risk of T2D, where improvements to follow-up and screening processes achieved by individualisation could be valuable. This is particularly clinically important in BPB women given their increased risk of developing T2D relative to other ethnic groups, and the globally low uptake of postpartum diabetes screening in non-white individuals [55]. Among individuals without T2D (non-cases), women with a history of GDM displayed a higher mean PRS than age- and BMI-matched females without GDM and compared to males. This could be because these individuals have increased propensity to develop T2D in the future, but have not yet developed it, or because they have undiagnosed T2D, for example, through missed screening tests postdelivery.

We used data-driven clustering approaches in 2 separate models to explore how PRS might associate with the T2D subgroups that are increasingly recognised as a route to developing stratified diabetes care. Despite the absence of biomarkers such as auto-antibodies and C-peptide, which are seldom measured in the routine primary care datasets we had available, we were able to reproduce previously described diabetes subgroups [2,16,17] in a previously uninvestigated population of BPBs. We showed heterogeneous distributions of a T2D PRS across clusters in a model whose membership was defined by clinical measures. PRS was lowest in the MARD cluster, concordant with our findings that it performed best in predicting the risk of T2D in people aged under 40 years, and the MOD cluster, suggesting that other polygenic influences (e.g., on body weight) may be more important. Our SIRD cluster was characterised by markedly raised serum triglycerides and is worthy of further exploration in studies of rare genetic variants. In the absence of clinical measures of insulin secretion such as C-peptide, we were unable to fully delineate the previously well-replicated SIDD cluster in the Clinical Model. However, our MD cluster may include individuals who are insulin deficient as it is characterised by low serum triglycerides (1.21 (0.43) mmol/L) in addition to high HDL (2.07 (0.25) mmol/L), and our CUD cluster likely also contains people with SIDD as its membership is characterised by high HbA1c and low serum triglycerides.

When we incorporated the PRS in an integrated cluster model with clinical features, we observed additional delineation of subgroups than our previous model, with emergence of clear clusters representing IRD and SIRD. This enhanced delineation was possible without biochemical measures of insulin secretion and resistance or diabetes autoantibody data. As such measures are rarely performed in routine primary care for people with diabetes (due to cost, performance, and interpretative challenges), these findings suggest PRS could be a pragmatic tool to aid clinical care if cheap genotyping chips become widely used in health systems. Interestingly, we were also able to delineate a probable SIDD cluster, well described in other studies of South Asians [23,56], on the basis of high polygenic susceptibility that was underpinned by gene variants associated with insulin secretion, in combination with supportive clinical features including high HbA1c, and low BMI and serum triglycerides. These findings are

supported by tendencies towards higher scores on previously reported beta cell PRS [37] and fasting glucose PRS [18]. This analysis showed that the diabetes subgroups identified using the integrated cluster model were associated with differential rates of progression to complications, which were not apparent in the clusters found by the Clinical Model, implying that addition of PRS to data-driven identification of T2D subgroups could provide an unexplored clinical tool to risk-stratify populations and target care.

This study has several limitations, including our need to impute missing data not present in health records and the lack of a replication cohort to externally validate findings. Our analysis of the progression to T2D after GDM was limited by the low uptake of postpartum diabetes screening that may have resulted in underdiagnosis of T2D and ascertainment bias. Across all of our analyses, it is likely that some individuals coded as having T2D actually have type 1 diabetes or rare monogenic forms of diabetes, although the absolute number of these and impact on overall findings is likely to be very small. It is also likely that undiagnosed T2D may be present in some of our controls, and this is expected to lead to a higher mean PRS in that group, but attenuation of any effect sizes of analyses investigating its predictive ability. Misclassification of type 1 and type 2 diabetes may have occurred in this real-world data set, although the small number of individuals diagnosed with T2D managed with insulin alone ($n = 14$) suggests the effect of this on our findings would be minimal. In the absence of C-peptide and auto-antibody data in our primary care data set, estimating misclassification rates is complex and beyond the scope of this study. While miscoding of T2D may also have impacted results, previous research has suggested this occurs in fewer than 2% of individuals [57], and we would similarly expect the effect of this on results to be small.

In conclusion, our T2D PRS, optimised in British South Asians, modestly enhances prediction of incident disease when combined with an established clinical risk tool compared to using the clinical tool alone, and particularly in young people. Additionally, the PRS has value in predicting the onset of T2D in a specific high-risk group, women affected by gestational diabetes. These findings could aid the personalisation of care of people at risk of T2D. The PRS also helps elucidate aetiologically different diabetes subgroups at diagnosis, in the absence of insulin secretion/resistance measures, and these differ in their association with future complications. The value of PRS in this context may assist effective stratification of care in the future. Our work provides important insight into the genetic risk in an ethnic group underrepresented in research but disproportionately affected by T2D, and has significant potential to be translated into clinical practice.

## Supporting information

**S1 Text. Additional methods.**
(DOCX)

**S2 Text. TRIPOD checklist.**
(DOCX)

**S1 Fig. PRSs constructed using the clumping and *p*-value thresholding (C+T) method or from the PGS Catalog.** AUC, area under the curve; GWAS, genome-wide association study; OR, odds ratio; PRS, polygenic risk score; SD, standard deviation.
(DOCX)

**S2 Fig. Effect of altering age cutoff for older and younger participants on NRI in risk of incident T2D analysis.** NRI, net reclassification index; T2D, type 2 diabetes.
(DOCX)

**S3 Fig. Elbow plot to identify optimal number of clusters in latent class analysis using BIC and AIC.** AIC, Akaike information criteria; BIC, Bayesian information criteria.
(DOCX)

**S4 Fig. Manhattan plot and Q-Q plot for the GWAS in G&H.** G&H, Genes & Health; GWAS, genome-wide association study.
(DOCX)

**S5 Fig. Distribution of previously reported glycaemic trait PRSs across clusters in the Integrated Model.** BMI, body mass index; CI, confidence interval; CIR, corrected insulin response; FINS, fasting insulin; HbA1c, hemoglobin A1c; IRD, Insulin-Resistant Diabetes; ISI, insulin sensitivity index; MARD, Mild Age-Related Diabetes; MOD, Mild Obesity-related Diabetes; PRS, polygenic risk score; pSIDD, Probable Severe Insulin-Deficient Diabetes; SIRD, Severe Insulin-Resistant Diabetes; T2D, type 2 diabetes.
(DOCX)

**S6 Fig. Cox proportional hazard models to show association between cluster membership and development of macro- and microvascular complications of T2D.** CUPS, Clinically Undifferentiated High Polygenic Susceptibility Diabetes; IRD, Insulin-Resistant Diabetes; MARD, Mild Age-Related Diabetes; MOD, Mild Obesity-related Diabetes; SIRD, Severe Insulin-Resistant Diabetes; T2D, type 2 diabetes.
(DOCX)

**S1 Table. Clinical codes used to define conditions.**
(XLSX)

**S2 Table. Characteristics of individuals used to construct T2D PRS in G&H.** G&H, Genes & Health; PRS, polygenic risk score; T2D, type 2 diabetes.
(XLSX)

**S3 Table. List of variants used in the T2D PRS for G&H.** G&H, Genes & Health; PRS, polygenic risk score; T2D, type 2 diabetes.
(XLSX)

**S4 Table. Three genome-wide significant variants associated with T2D in G&H.** G&H, Genes & Health; T2D, type 2 diabetes.
(XLSX)

**S5 Table. Six genome-wide significant variants associated with T2D that were reported by Kooner and colleagues [32].** T2D, type 2 diabetes.
(XLSX)

**S6 Table. T2D-associated loci transferable between Europeans and BPB individuals in G&H.** BPB, British Pakistani and Bangladeshi; G&H, Genes & Health; T2D, type 2 diabetes.
(XLSX)

**S7 Table. Results of the trans-ancestry colocalisation analysis to assess sharing of causal variants for transferable loci.**
(XLSX)

**S8 Table. Association between T2D PRS (T2D PRS) and selected variables included in analyses.** PRS, polygenic risk score; T2D, type 2 diabetes.
(XLSX)

**S9 Table. C-Index, categorical NRI, continuous NRI, and associated 95% CIs.** CI, confidence interval; C-Index, concordance index; NRI, net reclassification index.
(XLSX)

**S10 Table. Characteristics of participants after reclassification by IRT versus QDiabetes Model A alone.** IRT, integrated risk tool.
(XLSX)

**S11 Table. Number of participants reclassified as high (>10%) and low (<10%) risk of developing T2D over 10 years of follow-up by the IRT compared to QDiabetes Models A, B, and C.** IRT, integrated risk tool; T2D, type 2 diabetes.
(XLSX)

**S12 Table. Cox proportional hazard models describing association between T2D PRS and development of T2D in 302 women with a history of GDM.** GDM, gestational diabetes mellitus; PRS, polygenic risk score; T2D, type 2 diabetes.
(XLSX)

**S13 Table. Distribution of previously reported glycaemic trait PRS and wGRS across clusters in the Integrated Model.** PRS, polygenic risk score; wGRS, weighted genetic risk scores.
(XLSX)

## Acknowledgments

We thank Social Action for Health, Centre of The Cell, members of our Community Advisory Group, and staff who have recruited and collected data from volunteers. We thank the NIHR National Biosample Centre (UK Biocentre), the Social Genetic & Developmental Psychiatry Centre (King's College London), Wellcome Sanger Institute, and Broad Institute for sample processing and genotyping. We thank Barts Health NHS Trust, NHS Clinical Commissioning Groups (Hackney, Waltham Forest, Tower Hamlets, Newham), East London NHS Foundation Trust, Bradford Teaching Hospitals NHS Foundation Trust for GDPR-compliant data sharing. We also thank Sally Hull and Martin Sharp from the primary care data team at QMUL for their help in estimating population prevalence of type 2 diabetes. Most of all, we thank all of the volunteers participating in Genes & Health.

The Genes and Health Research team includes (in alphabetical order): Shaheen Akhtar, Mohammad Anwar, Elena Arciero, Samina Ashraf, Gerome Breen, Raymond Chung, Charles J Curtis, Maharun Chowdhury, Grainne Colligan, Panos Deloukas, Ceri Durham, Sarah Finer, Chris Griffiths, Qin Qin Huang, Matt Hurles, Karen A Hunt, Shapna Hussain, Kamrul Islam, Ahsan Khan, Amara Khan, Cath Lavery, Sang Hyuck Lee, Robin Lerner, Daniel MacArthur, Bev MacLaughlin, Hilary Martin, Dan Mason, Shefa Miah, Bill Newman, Nishat Safa, Farah Tahmasebi, Richard C Trembath, Bhavi Trivedi, David A van Heel, and John Wright.

## Author Contributions

**Conceptualization:** Hilary C. Martin, Sarah Finer.

**Data curation:** Sam Hodgson, Neneh Sallah, David A. van Heel, Hilary C. Martin, Sarah Finer.

**Formal analysis:** Sam Hodgson, Qin Qin Huang, Neneh Sallah, David A. van Heel, Rohini Mathur, Hilary C. Martin, Sarah Finer.

**Funding acquisition:** Chris J. Griffiths, David A. van Heel, Hilary C. Martin, Sarah Finer.

**Investigation:** Sam Hodgson, Neneh Sallah, R. Thomas Lumbers, Karoline Kuchenbaecker, David A. van Heel, Rohini Mathur, Hilary C. Martin, Sarah Finer.

**Methodology:** Sam Hodgson, Qin Qin Huang, Neneh Sallah, R. Thomas Lumbers, Karoline Kuchenbaecker, David A. van Heel, Rohini Mathur, Hilary C. Martin, Sarah Finer.

**Project administration:** Chris J. Griffiths, William G. Newman, Richard C. Trembath, John Wright, David A. van Heel, Hilary C. Martin, Sarah Finer.

**Resources:** Chris J. Griffiths, William G. Newman, Richard C. Trembath, John Wright, R. Thomas Lumbers, David A. van Heel, Hilary C. Martin, Sarah Finer.

**Supervision:** R. Thomas Lumbers, Karoline Kuchenbaecker, David A. van Heel, Hilary C. Martin, Sarah Finer.

**Validation:** Sam Hodgson, Neneh Sallah, Hilary C. Martin, Sarah Finer.

**Visualization:** Sam Hodgson, Qin Qin Huang, Neneh Sallah, Hilary C. Martin, Sarah Finer.

**Writing – original draft:** Sam Hodgson, Qin Qin Huang, Sarah Finer.

**Writing – review & editing:** Sam Hodgson, Qin Qin Huang, Neneh Sallah, Chris J. Griffiths, William G. Newman, Richard C. Trembath, John Wright, R. Thomas Lumbers, Karoline Kuchenbaecker, David A. van Heel, Rohini Mathur, Hilary C. Martin, Sarah Finer.

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
