## [Editor Report · Decision Letter 0]

26 Aug 2021

Dear Dr Finer, 

Thank you for submitting your manuscript entitled "Harnessing the power of polygenic risk scores to predict type 2 diabetes and its subtypes in a high-risk population of British Pakistanis and Bangladeshis in a routine healthcare setting" for consideration by PLOS Medicine.

Your manuscript has now been evaluated by the PLOS Medicine editorial staff as well as by an academic editor with relevant expertise and I am writing to let you know that we would like to send your submission out for external peer review.

Please re-submit your manuscript within two working days, i.e. by Aug 30 2021 11:59PM.

Kind regards,

Callam Davidson

Associate Editor

PLOS Medicine

---

## [Decision Letter · Decision Letter 1]

3 Nov 2021

Dear Dr. Finer,

Thank you very much for submitting your manuscript "Harnessing the power of polygenic risk scores to predict type 2 diabetes and its subtypes in a high-risk population of British Pakistanis and Bangladeshis in a routine healthcare setting" (PMEDICINE-D-21-03668R1) for consideration at PLOS Medicine. 

Your paper was evaluated by an associate editor and discussed among all the editors here. It was also discussed with an academic editor with relevant expertise, and sent to independent reviewers, including a statistical reviewer. The reviews are appended at the bottom of this email and any accompanying reviewer attachments can be seen via the link below:

[LINK]

In light of these reviews, I am afraid that we will not be able to accept the manuscript for publication in the journal in its current form, but we would like to consider a revised version that addresses the reviewers' and editors' comments. Obviously we cannot make any decision about publication until we have seen the revised manuscript and your response, and we plan to seek re-review by one or more of the reviewers. 

We hope to receive your revised manuscript by Nov 24 2021 11:59PM. Please email us (plosmedicine@plos.org) if you have any questions or concerns.

We look forward to receiving your revised manuscript. 

Sincerely,

Callam Davidson, 

PLOS Medicine

plosmedicine.org

Please revise your title according to PLOS Medicine's style. Your title must be nondeclarative and not a question. It should begin with main concept if possible. "Effect of" should be used only if causality can be inferred, i.e., for an RCT. Please place the study design ("A randomized controlled trial," "A retrospective study," "A modelling study," etc.) in the subtitle (ie, after a colon).

Abstract Background: Provide the context of why the study is important. The final sentence should clearly state the study question.

Abstract Methods and Findings:

* Please include the study design, years during which the study took place, and main outcome measures.

* Please quantify the main results (with 95% CIs and p values).

* Please include the important dependent variables that are adjusted for in the analyses.

Abstract Conclusions:

* Please address the study implications without overreaching what can be concluded from the data; the phrase "In this study, we observed ..." may be useful.

* Please avoid vague statements such as "these results have major implications for policy/clinical care". Mention only specific implications substantiated by the results.

Please add line numbering in the margins of the document to facilitate the review process.

Author summary bullet points 2 and 3: Please temper assertions of primacy (“No study has systematically…”) by adding ‘to our knowledge’ or similar. Please check throughout (other instances I noted were in the first and second paragraphs of the discussion). 

Please remove the final line of your introduction (“Our findings have major clinical relevance to a wider UK and global population of people of south Asian ancestry at risk of type 2 diabetes”). I would suggest toning this statement down and relocating it to the discussion.

Please remove funding information from the acknowledgements, in the event of publication this information is published as metadata based on your responses to the submission form questions.

For similar reasons, please also remove the conflict of interest and contributions statements from the end of the main text and ensure this information is captured as part of the submission form.

Please add ‘[preprint]’ after medRxiv for references 9, 13, and 41. 

Please ensure you define all abbreviations used in your tables.

In Figures 1-3, please show the axis beginning at zero. If this is not possible, please show a break in the axis.

Did your study have a prospective protocol or analysis plan? Please state this (either way) early in the Methods section.

Please consider adding an additional figure providing an overview of the study design (see Figure 1 in https://journals.plos.org/plosmedicine/article?id=10.1371/journal.pmed.1003498 for an example). This will aid comprehension for the non-specialist reader.

Please ensure that the study is reported according to the TRIPOD guideline, and include the completed TRIPOD checklist as Supporting Information. Please add the following statement, or similar, to the Methods: "This study is reported as per the Transparent Reporting of a Multivariable Prediction Model for Individual Prognosis or Diagnosis (TRIPOD) guideline (S1 Checklist)."

The TRIPOD guideline can be found here: https://www.equator-network.org/reporting-guidelines/tripod-statement/

Comments from the Academic Editor:

1. The diagnosis of diabetes was based on EMR (no OGTT was performed to screen for diabetes). This will have led to a low incidence of diabetes, and will have been largely affected by economic status.

2. The genetic risk score was calculated based on the findings from European-ancestry individuals, I would suggest to use a South Asian genetic risk score or simply perform a replication for all reported loci for T2D with samples from South Asian individuals first.

Comments from the reviewers:

Reviewer #1: The authors perform a wide range of analysis to explore the genetic architecture of type 2 diabetes (T2D) in British Pakistanis and Bangladeshis (BPB). First the authors perform a genome-wide association study (GWAS) and contrast results with a large GWAS of European ancestry individuals. Secondly, the authors construct a polygenic risk score specific for BPB and check added predictive performance over the existing clinical score QDiabetes.

My comments focus on the methodological details of these two main analysis steps.

1. Transfer of previously-identified GWAS loci to the study population

a) Please provide more details (e.g. citation, formulas or code) how the "expected power for replication" was calculated.

b) How did you calculate the expected proportion of genetic loci one might expect to replicate in a sample of a given ancestry given the power and linkage disequilibrium differences? Please provide more details (e.g. citation, formulas or code).

c) 76 loci are classified as "transferrable". Yet only 27 loci are analysed for colocalisation using TEColoc. Please explain why 49 loci are not considered. Is this due to the filter that ≥10% overlapping variants should be present? If so, please explain why there is so little overlap between Vujkovic et al European-ancestry GWAS and the BPB G&H GWAS.

d) Regarding imputation, in the Supplementary File 1 the number of SNPs used to construct PRS (N=9,527,863) is provided, but not the number for GWAS or colocalization analysis. Please provide these missing two numbers of SNPs as well. Why did you use different set of SNPs for different analysis?

e) Please provide more details on the TEColoc or the JILM method and why you selected this method in contrast to the more commonly used colocalisation method by Giambartolomei et al.

2. Development of an integrated risk tool to predict incident type 2 diabetes

a) The authors show two ways to calculate the categorical net reclassification index (NRI): 

(1) the proportion of individuals who subsequently developed type 2 diabetes correctly reclassified as high risk, minus the proportion of individuals who subsequently developed type 2 diabetes incorrectly reclassified as low risk; 

(2) the proportion of individuals who did not develop type 2 diabetes correctly reclassified as low risk, minus the proportion of individuals who did not develop type 2 diabetes incorrectly reclassified as high risk.

Please be more precise how the proportions are defined, what exactly is the numerator and the denominator of the proportion?

b) Which of the two definitions was used for Figure 1 and in the Results?

c) Figure 1 shows the NRI for all subjects and cases and non-cases. How can the "proportion of individuals who subsequently developed type 2 diabetes" be calculated for non-cases?

d) There is a big distinction in the pattern of NRI for subjects younger or older than 40. How robust are the results with respect to this cut-off?

Minor:

- Page 5: "with descriptive statistics. ." Please remove the second full period.

Reviewer #2: In this paper, Hodgson et al. construct polygenic risk scores and integrate them with clinical models to enhance the prediction of type 2 diabetes (T2D) in UK Pakistani and Bangladeshi individuals in a large population study. They first assess the transferability of known T2D genetic risk loci (derived from predominantly European populations) to their cohort and replicated significantly fewer loci than predicted. They then constructed a number of polygenic scores using variants identified from a multi-ancestry GWAS and adjusted/tuned the scores for Pakistani and Bangladeshi individuals in 2249 cases and 4443 controls, selecting the one with the highest odds for prediction. This score was then integrated with three versions of a clinical model (QDiabetes) to predict onset of T2D over 10 years in 13,642 individuals free from diabetes at baseline. The authors show that incorporation of the PRS with the QDiabetes provided better discrimination of risk of progression to T2D, especially in those developing T2D <40 years. The PRS also was noted to be higher in women with a history of gestational diabetes who progressed to T2D versus non-progressors. Finally, they incorporated the PRS into a cluster analysis of baseline routine clinical characteristics, first largely replicating clusters defined previously in white populations, but also defining a 'clinically undifferentiated' cluster, associated with higher PRS but with no relationship to risk of progression to microvascular and macrovascular complications.

This study has several strengths:

1. The analysis was conducted in a large population-based cohort of a hitherto understudied ethnic group; UK Pakistani and Bangladeshi individuals, who are at a disproportionately higher risk of developing T2D even within the south Asian ethnic group and at relatively lean body mass. It would be reasonable to postulate that this underlying risk is, in part, mediated by genetic predisposition.

2. Longitudinal data linkage of participants to their electronic health record further provides a very powerful methodology in which genetic data can be associated with clinical outcomes. There are very few longitudinal cohorts of this size with linked data.

3. This study uses multimodal approaches to decipher phenotype-genotype associations, including cluster analyses to identify heterogeneity in T2D sub-phenotypes, which is relatively novel in this ethnic group.

On the other hand, there are some issues that must be addressed to enhance the relevance and focus of the paper:

1. The authors chose to apply polygenic scores derived in predominantly white European ethnicity to their Pakistani and Bangladeshi population and 'fine tuned' the effect size of variants to enable the best prediction of T2D. If their only goal had been to examine the transferrability of European PRS to South Asians, this may have been justified; however, their stated goal in the first sentence of the Discussion (also the more clinically relevant one) is to improve the prediction of T2D in South Asians via genetic means, and in that case the genetic predictive tool should include variants discovered in this population. One alternative, somewhat hindered by sample size, would have been to leverage the GWAS performed on cases and controls in this specific G&H cohort, and incorporate those variants into the score. Arguably, higher rates of consanguinity in this ethnicity have the potential to uncover novel loci. Nevertheless, at a minimum, the variants discovered by Kooner et al. (Nature Genetics 2011) should have also been included, as the trans-ancestry GWAS used to identify T2D loci has a very small number of South Asian individuals. The decision not to include South Asian-specific variants should be discussed or justified.

2. In this vein, the GWAS conducted in G&H should be at least briefly presented, including the standard Manhattan and Q-Q plots and a table of the genome-wide significant associations, and placed in the context of the prior Kooner et al. GWAS.

3. One other major limitation is the absence of C-peptide or fasting insulin in this cohort, which pre-empts the ability to generate clusters analogous to those described by Ahlqvist et al. (ref. 19). Since this is an essential marker to denote beta-cell function, the authors are unable to reconstruct the "severe insulin deficient diabetes" (SIDD) cluster originally reported by Ahlqvist et al. Indeed, it is curious that what they term a novel "clinically undifferentiated" cluster has the strongest association with the T2D PRS. This is very telling: a close look at Supplementary Table 4 reveals that the PRS constructed here is largely composed of the variants that have the strongest effect sizes on T2D (i.e. those that were originally discovered in Europeans but replicate in the current smaller sample). Therefore they are the ones that emerged from the earliest European GWAS, which by design favored loci that caused defects in beta-cell function (e.g. TCF7L2, MTNR1B, SLC30A8, KCNJ11, HNF4A, CDKAL1, IGF2BP2; see Pubmed ID 18504548 for an in-depth discussion of how the early T2D GWAS yielded beta-cell genes). Thus it is quite possible that the "clinically undifferentiated" cluster might indeed correspond to a cluster of severe, genetically driven beta-cell dysfunction, analogous to the SIDD cluster of Ahlqvist et al. As it turns out, a genetic exploration of the Ahlqvist clusters already available in preprint form (https://www.medrxiv.org/content/10.1101/2020.09.29.20203935v1) does demonstrate the strongest association of the T2D PRS with the SIDD cluster.

4. The authors do not discuss a clear limitation that the time to follow-up of 10 years does not conclusively distinguish cases from controls, in that some participants classified as controls will eventually develop diabetes. Therefore, scores in the control group, given the high rates of T2D in this population are likely to be higher than true diabetes-free controls. Indeed, the mean age of participants in the dataset used to construct scores was 30.8 years (supp Table 2). Could they not have chosen 'older' diabetes free controls? Further the controls used to construct polygenic scores are described as having 'insufficient data to be included in the study' - it is not clear what is meant by this.

5. The authors acknowledge that there may be miscoding of T2D due to misclassification of diabetes as a limitation, but to identify cases, it appears they relied only on the first appearance of the diagnostic code for T2D in the electronic health record. Could they confirm whether this was verified for example, through confirmation of a repeat diagnostic code at a subsequent encounter, prescription data and if not, could they discuss how likely this is to have been an issue in real-world electronic health records.

6. What might the effect of a highly consanguineous population be on generation and efficacy of a PRS. The authors say they selected 'unrelated' cases and controls - however, should not a PRS in this specific population take into account high proportions of relatedness? Could the genomic data be used to exclude a given degree of relatedness, measured as "identity by descent" in available software such as PLINK? 

7. There are likely to be a number of reasons why fewer loci were replicated including differences in minor allele frequency, linkage disequilibrium, gene-environment interactions or the simple statistical phenomenon of the "winner's curse" (an overestimate of the effect size reported in the initial study) - a more extensive discussion of this might have been useful.

8. For the GDM analysis, the sample sizes used to compare PRS between progressors and non-progressors are small. Could the authors verify there was sufficient power based on expected effect sizes from previous studies?

9. Models B and (less so) C that incorporate glycemic measures (glucose and HbA1c, respectively), attenuate the ability of the PRS is predicting T2D onset, suggesting that these biochemical measures probably identify those at risk of T2D. This undermines the utility of the PRS, and confirms previous findings from multiple studies that polygenic prediction of T2D does not add much to clinical predictors if glycemia is included in the instrument (see, e.g. Meigs et al. and Lyssenko et al., both in NEJM 2008). The Lyssenko paper did show that genetic prediction was most useful in younger individuals or those with the longest follow-up, indicating that clinical predictors may not manifest themselves fully until later in life or closer to disease onset, a finding that is consistent with the current study. Since there was no relationship between PRS and incidence of complications, the counter-argument of genetic prediction lacking clinical utility could perhaps be discussed more critically?

10. There are multiple statistical comparisons (e.g. all the SNPs taken forward into this cohort), but there were no adjustments to the p-values for multiple comparisons, as a nominal p-value of 0.05 was taken as evidence of replication. While this is sensible when one is trying to estimate how many SNPs will reach that alpha level over the predicted number based on effect size and sample size, this should be acknowledged and explained.

11. The cluster analysis seems 'tacked on' rather than being integral to the core work of PRS transferrability. As such, there is no discussion of the fact that the PRS is associated with different clusters in this analysis but few significant associations with SNPs were found in the original analysis by Ahlqvist et al. The more extensive genetic evaluation undertaken by that group and published in the preprint cited above provides a potential blueprint of what could be done here with the GWAS data obtained in G&H. There should have been some reference to the recent Indian study that undertook clustering and identified a novel insulin-deficient but non-autoimmune cluster (Anjana et al. Pubmed ID 33527813). 

12. This is a small point, but the Ahlqvist et al. study preceded and provided the reason for the subsequent Dennis et al. study - since they are cited at the same place in the manuscript, they should be listed in chronological order.

Reviewer #3: The study by Hodgson et al aims at developing a PRS to estimate the 10-year risk of developing T2D and to identify T2D subtypes in patients of South Asian ancestry living in the UK. The authors tested the transferability to South Asians, of genomic variants associated to T2D in people of European descent and then tested the predictive ability of a more South Asian-specific PRS that they combined with the clinical score QDiabetes to estimate the risk of developing T2D in women with gestational diabetes and in men and women, younger or older than 40 years of age. 

Strengths: 

The manuscript is well written, and results are clear. 

The research question is important. People of South Asian origin are more affected by T2D. This occurs at a younger age and at lower BMI. The question that there may be a genetically-distinct T2D subtype in this population is sound. Most of PRS have been developed with and for populations of European descent. The transferability of data from Europeans to people of South Asian origin is unknown. 

The authors have access to a large set of clinical and genomic data including electronic health records and follow-up of British Pakistani and Bangladeshi (BPB). 

The analyses are appropriate and carefully done. 

The PRS may be useful for women with history of GDM. Women who subsequently developed T2D had a higher PRS than those who did not. Women with GDM had also higher PRS than age- and BMI-matched controls without GDM and higher than men. This clinically important observation could have been investigated further. 

Weaknesses. 

The construction of the PRS is not detailed enough. The number of SNPs that have been included and their effect size should be stated. How many of the European transferable SNPs have been included in the PRS?

Some of the conclusions are not well supported by the results. For instance, the predictive performance of the PRS over that of the QDiabetes score is very modest. Are the differences in AUC (Table S5) significant after the addition of the PRS? They seem to be significant only for Model A. The incremental effect of the PRS is eliminated by the addition of fasting glucose (Model B) or HbA1c (Model C) to the clinical score. 

Same comment could be made for the NRI that is significant only for Model A which has the lowest predictive performance. 

Data-driven clustering: T2D cluster driven by the PRS is the largest in term of number of individuals. However, the progression of microvascular and macrovascular outcomes in individuals of this cluster does not differ from the other clusters. The only cluster that differs between macro and microvascular complications is the MARD one. This could be explained by the fact that age has a different impact on the progression of macrovascular and microvascular outcomes. 

The cluster analyses could be done also individuals of European descent and the radar plots compared between the two ethnic groups. 

Minor: 

The title should be more specific to the findings.

[LINK]

---

## [Decision Letter · Decision Letter 2]

8 Feb 2022

Dear Dr. Finer,

Thank you very much for submitting your revised manuscript "Integrating polygenic risk scores in the prediction of type 2 diabetes risk and subtypes: an analysis of genomic and routine primary care health data in a population-based study of British Pakistanis and Bangladeshis" (PMEDICINE-D-21-03668R2) for consideration at PLOS Medicine. 

Your paper was sent back to independent reviewers and the reviews are appended at the bottom of this email. Any accompanying reviewer attachments can be seen via the link below:

[LINK]

As there are remaining concerns to be addressed, we will not be able to accept the manuscript for publication in the journal in its current form. We would however like to consider a revised version that addresses the reviewers' and editors' comments. As stated previously, we cannot make any decision about publication until we have seen the revised manuscript and your response, and we plan to seek re-review by one or more of the reviewers. 

We hope to receive your revised manuscript by Mar 01 2022 11:59PM. Please email us (plosmedicine@plos.org) if you have any questions or concerns.

We look forward to receiving your revised manuscript and please don't hesitate to get in touch if you have any questions. 

Sincerely,

Callam Davidson, 

Associate Editor

PLOS Medicine

plosmedicine.org

The current title is slightly too long. Please update to 'Integrating polygenic risk scores in the prediction of type 2 diabetes risk and subtypes in British Pakistanis and Bangladeshis: a population-based cohort study', or similar.

Thanks for providing a completed TRIPOD checklist - please update the checklist to use section and paragraph numbers as opposed to page numbers (as these change during the revision process). 

Please also ensure the TRIPOD checklist is cited in the methods (e.g. 'This study is reported according to the Transparent reporting of a multivariable prediction model for individual prognosis or diagnosis (TRIPOD) statement (S1 checklist)'. 

Comments from the reviewers:

Reviewer #2: The authors have been very responsive to the review process and the manuscript is significantly improved. Particularly I appreciate the new designation of the pSIDD cluster. There is a small number of minor outstanding issues that should be relatively straightforward to resolve.

1. The choice of SNPs to construct the most informative polygenic score: thank you for the clarification, and it does make sense to use the multiethnic resource that includes more samples than Kooner et al. I still think that some mention of the rationale used and elegantly outlined in the response to the reviewers should find its way into the manuscript, if anything to allay the concern of the informed reader. This could be done in the Supplement or as a couple of sentences in the Methods.

2. Multiple testing correction for SNPs taken forward: I agree that for a replication of a single variant, a P-value <0.05 is sufficient. When one is testing replication of multiple individual variants, the number of hypotheses is as large as the number of uncorrelated variants taken forward: thus strictly speaking it does make sense to use a Bonferroni correction, and this is done routinely. You could simply state that the variants passed the nominal P-value, but that experiment-wide significance would be set at 0.05/n.

3. As minor edits, in Supplementary Table 4, the locus "CDKAL1" has a typo ("CKDAL1"). Gene names should be italicized throughout, including Supplementary Tables

Reviewer #3: The modifications made to the revised version of the manuscript clearly add to the quality of this paper, and most of my concerns from my previous review have been addressed adequately. 

Reviewer #4: The authors explore the genetic determinants of Type 2 Diabetes (T2D) in people of British Pakistani and Bangladeshi (BPB) ancestry. Using genetic data from the Genes & Health project, they first explore whether SNPs associated with T2D in European populations are also associated with T2D in their BPB sample. They then construct a polygenic risk score (PRS) for T2D and show that it can be useful for predicting which individuals will develop T2D alongside an already established clinical tool (QDiabetes).

I would like to commend the authors for working with a previously understudied population group, and I agree with them that further genetic research involving such groups is needed.

As a statistician, I am perhaps unable to fully scrutinise the authors' applied results, but I will attempt to provide some comments on their use of appropriate methodology to obtain these results. Since I was not involved in the previous round of review, some of my comments will be complementary to those previously raised by other reviewers, while other comments will be different. 

Here are my comments on the manuscript.

1) The authors used a European LD panel (European 1000G samples) for their LD clustering and thresholding analysis. Was it not possible to use a south Asian panel instead?

2) In their supplement, the authors present a formula according to which they calculated the proportion of SNPs from the European GWAS that would be expected to replicate in the G+H GWAS. First of all, the formula seems wrong; for example, in a case-control study with n_cases = n_controls, one would have 1 - phi = 0 and hence only a 0.05 probability to replicate a SNP regardless of its beta coefficient in the European GWAS. Please clarify (could it be that phi = n_cases / n instead?). Moreover, the authors should provide an explanation about why the formula does what they are trying to achieve, or a reference to support its use. In general, I found it a bit surprising that 72% of these GWAS-significant variants should be expected to not associate with T2D even at a nominal 0.05 level in the authors' genetic sample.

3) In the authors' colocalization analyses, the overlap of genetic variants between the European and G+H GWAS in each region was as low as 10%. Therefore, it is quite likely that in some regions the truly causal variant(s) may not have been genotyped in one of the two samples (or even in none of the two samples). Would that be a problem for the TEColoc method used by the authors?

4) What was the explanatory power (e.g. in terms of R^2 values) of the diabetes PRS constructed by the authors? Also, did the PRS associate with any of the other variables included in their analysis (BMI, HDL, TG, HBA1c, glucose)? Following from a previous comment by another reviewer, any associations with glucose and HBA1c would be particularly interesting to report, since they would explain why the PRS offered little benefit over QDiabetes when using models B or C which incorporated these traits.

5) To assess the predictive accuracy of their PRS, the authors use the net reclassification index (NRI), which amounts to first binarizing each method's predictions by splitting individuals to high-risk and low-risk groups and then comparing the proportions of cases in each group. Wouldn't it make more sense to not binarize and instead compare each method's predictive accuracy using the actual estimated probabilities? E.g. something like Sum( |Pr(case) - Case.status| ) where Case.status = 1 for cases and 0 for controls, and the summation extends over all individuals.

6) One possible limitation of the study is the potential for its results to be affected by selection bias. For example, since BPB individuals were only recruited at three centres across the UK (East London, Bradford and Manchester), some of their baseline characteristics may differ to those of other such populations. The construction of the PRS was based on an even more selective sample, since individuals included in the QDiabetes assessment sample were excluded from the PRS construction. In addition, genetic studies conducted on individuals with a pre-established disease are known to suffer from a form of selection bias called index event bias; this may have been present in the authors' analysis on T2D incidence in women who had previously developed gestational diabetes.

7) As mentioned by the authors, one of the main reasons for studying T2D in BPB individuals is the increased frequency of diabetes in this population group. Can any of the authors' findings help to explain why T2D is more common in BPB individuals? 

8) In their conclusions, the authors state that the addition of their PRS to data-driven identification of T2D subgroups can improve the classification of individuals into T2D subgroups. It is perhaps worth clarifying that this seems to be the case only if information on other relevant variables is missing. In general, the results of a statistical clustering algorithm depend on what variables the clustering was based on. Presumably, the authors' PRS was only needed to identify and characterize the pSIDD cluster because information on fasting glucose and beta cell function-related traits was not available in their dataset, and the PRS would be redundant if such information was available.

[LINK]

---

## [Decision Letter · Decision Letter 3]

24 Mar 2022

Dear Dr. Finer,

Thank you very much for re-submitting your manuscript "Integrating polygenic risk scores in the prediction of type 2 diabetes risk and subtypes in British Pakistanis and Bangladeshis: a population-based cohort study" (PMEDICINE-D-21-03668R3) for review by PLOS Medicine.

I have discussed the paper with my colleagues and the academic editor and it was also seen again by two reviewers. I am pleased to say that provided the remaining editorial and production issues are dealt with we are planning to accept the paper for publication in the journal.

[LINK]

We look forward to receiving the revised manuscript by Mar 31 2022 11:59PM.   

Sincerely,

Callam Davidson, 

Associate Editor 

PLOS Medicine

plosmedicine.org

Requests from Editors:

Please trim your Author Summary, aiming for 2-3 single sentence bullet points per question (I think you can provide slightly less detail for the ‘Why was this study done?’ question). Please include the headline numbers from the study, such as the sample size and key findings, under the ‘What did the researchers do and find?’ question.

Please define any abbreviations used in your Tables and Figures in the associated legend. 

Your line numbering should be continuous throughout the document rather than beginning again on each page. 

Page 5, Line 30: Please cite Table 1 in the Results rather than the Methods section.

Please consider using a different colour scheme for your Figures and Supplementary Figures to aid those with red/green colour blindness. 

The ordering of your Supporting Figures is non-sequential (Figure S4 is cited before S2 and S3). Please rearrange the figures such that they are cited sequentially. 

Please define the 95% CI bars in the legend of Figure 3.

Page 7, Line 47: Should this be Figure 3B-C rather than A-B?

In your Supplementary Tables, please report p values as P<0.001. 

It appears the x-axis label in Figure 4 has been cut off.

Page 8, Lines 23-25: This feels like Discussion material (speculative) – please consider relocating.

Page 8, Lines 45-50: Similar to the above, this feels like Discussion material. Please keep to reporting findings in the Results and add interpretation only in the Discussion. 

I could not find reference to Figure S5 in the main text.

Page 10, Line 9: ‘To our knowledge’.

Comments from Reviewers:

Reviewer #2: The authors ought to be commended for their thorough and responsive approach to the review process. I have no further comments.

Reviewer #4: I am happy with the authors' response to my previous comments and would like to recommend that their manuscript is accepted for publication.

[LINK]

---

## [Editor Report · Decision Letter 4]

6 Apr 2022

Dear Dr Finer, 

On behalf of my colleagues and the Academic Editor, Dr Weiping Jia, I am pleased to inform you that we have agreed to publish your manuscript "Integrating polygenic risk scores in the prediction of type 2 diabetes risk and subtypes in British Pakistanis and Bangladeshis: a population-based cohort study" (PMEDICINE-D-21-03668R4) in PLOS Medicine.

When making the formatting changes, please also ensure the institutional affiliation of author NS is up to date (the competing interests statement suggests it may not be). 

PRESS

Sincerely, 

Callam Davidson 

Associate Editor 

PLOS Medicine